# Predicting Return to Work Following Myocardial Infarction: A Prospective Longitudinal Cohort Study

**DOI:** 10.3390/ijerph19138032

**Published:** 2022-06-30

**Authors:** Weizhe Sun, Leila Gholizadeh, Lin Perry, Kyoungrim Kang

**Affiliations:** 1School of Nursing and Midwifery, Faculty of Health, University of Technology Sydney, Ultimo 2007, Australia; weizhe.sun-1@alumni.uts.edu.au (W.S.); leila.gholizadeh@uts.edu.au (L.G.); lin.perry@uts.edu.au (L.P.); 2College of Nursing, Research Institute of Nursing Science, Pusan National University, Yangsan 50612, Korea

**Keywords:** cardiac rehabilitation, modifiable factors, myocardial infarction, quality of life, return to work

## Abstract

This study aimed to determine the proportion of patients who returned to work within three months post-myocardial infarction and the factors that predicted return to work. A total of 136 participants with myocardial infarction completed the study questionnaires at baseline and three months post-discharge between August 2015 and February 2016. At the three-month follow-up, 87.5% (*n* = 49) of the participants who were working pre-infarction had resumed work. Age, gender, education, smoking, readmission after discharge, number of comorbidities, diabetes, social support, anxiety, and depression were significantly associated with returning to work at three months post-discharge. Age, gender, smoking, anxiety, and depression significantly predicted those patients with myocardial infarction that returned to work, using binary logistic regression. The majority of patients in work who experience myocardial infarction have the capacity to achieve a work resumption by three months post-discharge. Interventions that facilitate returning to work should focus on modifiable risk factors, such as improving these patients’ mental health, comorbid conditions, risk of readmission, smoking, and social support. Healthcare providers should work in partnership with patients’ family members, friends, and employers in developing and implementing interventions to address these modifiable factors to facilitate patients’ return to work.

## 1. Introduction

Myocardial infarction (MI) is a life-threatening disease. In the UK, more than 100,000 people are admitted to hospitals because of MI every year, representing one admission every five minutes [1]. In the US, a person experiences MI about every 40 s, and about 14% of patients experiencing MI die as a result within the first year [2]. In 2018, around 7300 people lost their lives to MI in Australia, the equivalent of an average of 20 deaths per day [3]. In South Korea, mortality from coronary heart disease (CHD), including MI, rose enormously from 1983 to 2012 [4]. Nevertheless, the death rates from MI have improved significantly over the last couple of decades in high-income countries because of the improved health systems and implementation of effective public health strategies [5]. For example, in Australia, the age-standardized MI incidence declined from approximately 377 to around 302 per 100,000 between 2013 and 2018 [6]. The improvements in the mortality rates have been mainly due to the use of effective evidence-based approaches in the diagnosis, treatment, and management of MI [5]. Patients with MI are now living longer than in previous years. Whilst this is a significant achievement, this may not necessarily indicate that survivors of MI live with a high quality of life (QOL) [7]. The physical, psychological, financial, and social aspects of these patients’ lives are commonly adversely affected by the MI, by its treatment, the associated complications, and the need for lifestyle changes [7]. These considerations further negatively affect the QOL of patients with MI, and were reflected in a recent systematic review which indicated that patients with an MI who were less involved in physical activity, perceived higher anxiety, depression, and stress, and lower social support, and had a lower monthly income and reported a worse QOL than their counterparts [8].

Patients with MI who return to work (RTW) have reported a higher QOL than those who did not [9]. An RTW is one significant measure of patients’ cardiac rehabilitation (CR) success post-MI, and whether they recover fully from their illness and return to their normality [10]. An RTW post-MI may also indicate individuals with superior health post-MI, and an RTW itself may result in better health and financial statuses. In previous studies, people who were out of work were more likely to experience financial hardship, which may affect their medication affordability, and both their physical and mental health. They had an increased risk for premature death, CHD, depression, and anxiety compared to employed individuals [9,10]. However, the RTW rates post-MI have remained consistent, despite substantial improvements in the management of CHD in recent decades [11]. The rates reported for an RTW post-MI in low- and middle-income countries are much lower than those of higher-income countries. At one-year post-MI, only about 55% of participants had noted an RTW in a Chinese study [12], compared with the RTW rates of more than 90% in western countries [9,13]. Therefore, there is a need for further study on the topic to better support patients with MI with this crucial outcome.

### Review of the Literature

An RTW is an important step in a patient’s recovery journey. Patients who gradually resume their work show quicker recovery rates from their disease, indicating that they are able to return to a normal life and experience less associated financial and psychological effects [14]. In contrast, long-term work absence, work disability, and unemployment due to disease may negatively impact patients’ health and wellbeing [14]. Irrespective of the underlying medical conditions, the probability of an RTW has been seen to decline in patients who delay their work resumption, with the likelihood of an RTW dropping to 50% at 12 weeks of work absence among manufacturing workers [15]. Patients who did not RTW within six months post-MI were reported as unlikely to do so later [16], with only about 50% of participants with MI having the capacity to work at a two-year follow-up [17].

Although MI is an acute event, the effects of the disease on the patient and family can be long-lasting and profound. Apart from the acute physical effects, MI may affect the patients’ long-term psychological health. They may feel extreme anxiety and fear of another MI or death [18,19]. Due to this fear, patients may decrease their physical activity and social interactions [18]. In some patients, anxiety can be a result of the chest pain that they continue to experience after having an MI [20]. Some patients may feel powerless due to their perceived inability to manage their condition [20]. In addition, many patients feel fatigued and cannot trust their bodies as before [18,19,20]. These feelings can have a significant impact on the patients’ daily life; for instance, they may not be able to do their housework, socialize with family and friends, or pursue their hobbies as before [20]. These long-term emotional and lifestyle changes may adversely affect patients’ mental health, resulting in depression and anxiety, thus indirectly affecting the patients’ decision to RTW following MI.

To improve the outcomes and QOL of patients following MI, and to prevent recurrent MI episodes, CR is necessary. CR programs focus on developing patients’ skills in managing their physical and mental status and their cardiovascular risk factors via effective education and counseling [21]. Vocational evaluation and counseling have been identified as an essential part of CR to evaluate if an RTW is safe and feasible for the patient and to facilitate the RTW process [16]. A discussion about work-related issues should be held as early as possible, for example, before the patient is discharged, as this may provide the patient with a framework of expectations about an RTW [16]. Through counseling, healthcare providers are able to identify patients’ goals for an RTW and evaluate their workplace’s demands and concerns, working with patients to overcome barriers to RTW and facilitate work resumption. Among patients with MI after an RTW, recent research showed that people who took longer to achieve a work resumption and who participated in CR perceived less stress from work and a higher QOL [8,22,23]. Behavioral factors are also significantly associated with QOL among patients with MI after an RTW, which people who had better health behaviors, such as more health responsibility and exercise, perceived a better QOL [22].

Literature from western developed countries has explored the factors associated with patients’ RTW following MI. Studies from the US found that married people with greater physical health and no history of CHD and hypertension were more likely to RTW at 12 months post-MI [10], whereas hospital readmissions, smoking, and hypertension inhibited work resumption during this time [9]. A systematic review demonstrated demographic, behavioral, clinical, and psychosocial factors significantly related to a work resumption post-MI, including gender, age, educational level, work type, duration of hospitalization, comorbidities, complications, mental health, and self-evaluated general health [24]. However, no study has focused on the Korean population regarding this topic.

## 2. Materials and Methods

This study is a secondary analysis of data from a larger primary study that aimed to assess patients’ QOL post-MI and examine those clinical, behavioral, and psychosocial factors that predicted their QOL at three months post-MI [22]. In the current study, we specifically focused on an RTW as an important outcome, which is related to QOL in patients post-MI.

### 2.1. Aims

The aim of the current study was to determine the characteristics and proportion of patients who RTW within three months post-MI, and the predictors of this outcome. The research questions were:(1)What proportion of patients with myocardial infarction returned to work by three-months post-discharge in South Korea?(2)What demographic, behavioral, and clinical factors predicted these patients’ return to work?

### 2.2. Design

The original study selected a prospective longitudinal cohort design [25], as the same participants were assessed at baseline and followed over three months [26]. As part of the original study, data on the RTW status of the participants were collected at the three-month follow-up, and these data were analyzed in the current study. The STrengthening the Reporting of OBservational studies in Epidemiology (STROBE) checklist was followed to ensure rigor and transparency in the current study (Appendix A).

### 2.3. Participants

The setting of the original study was two tertiary hospitals with integrated cardiovascular centers located in the southern area of South Korea. Participants were recruited from the inpatient cardiovascular wards within a few days of admission to the hospitals with a diagnosis of MI. Patients were eligible for inclusion in the study if they (1) had a diagnosis of MI by a cardiologist; (2) had adequate Korean language skills to complete the study questionnaires; (3) resided in South Korea; (4) demonstrated the ability to understand the study and provide written informed consent. Participants were excluded if they: (1) had cognitive impairment; (2) were participating in other interventional studies at the time when the original study was being conducted. In total, 215 patients who met the inclusion criteria were screened between August 2015 and February 2016. Of these, 41 patients were excluded due to inadequate hearing and poor health conditions, and 24 refused to participate. Finally, 150 (69.8%) of those who were eligible to participate in the original study were recruited. Participants with regular employment for a definite and more or less extended period of time were regarded as the employed group. Casual workers were those who were paid daily on the basis of hours worked. To make the study outcomes measurable and clear, we only considered paid work as employment, and excluded unpaid work, such as household activities, volunteer work, or unpaid family work.

Participants who did not work prior to their MI were also included in the analyses, as they might still have the ability to start a job after their recovery from MI. This study did not exclude patients if they reached the statutory age of retirement, which is 60 years in South Korea [27], as it is common for Korean people to work beyond this age. In 2016, 45% and 33% of South Korean individuals within the age groups of 65–69 and 70–74 years were employed, respectively, by comparison to an average of 26% and 15% among all Organization for Economic Co-operation and Development (OECD) countries [28]. In a national employment survey of older workers conducted in South Korea in 2017, almost two-thirds of those aged 55–79 years worked because they needed to earn their living [28]. The effective retirement age in South Korea was reported at about 72 years for both men and women in 2016, much older than the average age within the OECD countries, at 65 and 63 years for men and women, respectively [28].

### 2.4. Data Collection

Before commencing data collection, potential participants were screened using the study inclusion and exclusion criteria. Participants were invited to the original study after they had received an explanation of the purpose and protocol of the study. They were provided with an information statement and signed the study consent form. Baseline data were collected from participants between August 2015 to February 2016, and the follow-up data were collected three months after discharge; 136 participants completed the follow-up questionnaires (Figure 1). Data were collected using questionnaires that comprised general characteristics, mental health status, and participants’ social support. The baseline assessments were collected within a few days after the participants were admitted to the hospitals with a diagnosis of MI, and the three-month follow-up assessments were carried out via telephone or face-to-face when participants visited the hospital for their usual follow-up care. The timeframe of three months was decided in consultation with two cardiologists from the hospitals, in consideration of the transition of patients after MI from the acute phase to a more stable condition [25].

### 2.5. Data Collection Instruments

#### 2.5.1. Demographic, Behavioral, and Clinical Data

To answer research question 2, demographic, behavioral, and clinical variables were chosen based on the findings of a systemic review of factors related to RTW after MI [24].

The demographic profiles were gathered from the participants using a questionnaire that included questions on age, gender, marital status, educational level, current employment status, and perceived financial situation. Health behavior profiles were collected, including participants’ physical activity status, average sitting time per day, smoking, and alcohol consumption.

A question was asked regarding the participants’ employment status. RTW, in the current study, was defined as working as a regular, casual, or self-employed worker. If the participant was an unpaid family worker or retired/unemployed post-MI, they were considered as not an RTW.

Clinical profiles were collected from the participants’ medical records, including their height, weight, diagnosis of MI, left ventricular ejection fraction (LVEF), and medical history, including hypertension, diabetes, hypercholesterolemia, cancer, stroke, mental health, and other heart diseases, time since the first diagnosis of heart disease, previous MI, time since the previous MI, and the type of interventions for the index MI.

At the three-month follow-up, patients’ experiences post-MI, including readmission to hospital, frequency of visiting a doctor, and frequency of chest pain within three months post-discharge, were also collected.

#### 2.5.2. Depression, Anxiety, and Stress Scale

The Depression, Anxiety, and Stress Scale (DASS-21) was used to assess the participants’ mental health status. This tool has the advantage of being less burdensome to respondents, allowing a concurrent assessment of three major psychological symptoms; high validity and reliability have been demonstrated in various population groups [29], including people with cardiovascular disease [30]. A Korean version of the DASS-21 is available and has been validated among the Korean population [31,32].

#### 2.5.3. The ENRICHD Social Support Inventory

The Enhancing Recovery in Coronary Heart Disease (ENRICHD) Social Support Inventory (ESSI) was used to evaluate the participants’ perceived social support. This scale has displayed good internal consistency, including a Cronbach’s alpha of 0.86 in patients with MI and 0.88 in patients post PCI [33,34]. The ESSI also has been said to have better convergent validity and discriminant validity compared with other social support measurement tools [33].

### 2.6. Ethical Considerations

The original study received ethical approval from the participating hospitals in South Korea (IRB no. H-1505-008-029; IRB no. 05-2015-072) and then obtained ratification from the Human Research Ethics Committee of the relevant university (HREC Approval No. 2015000254). The current study received ethical approval from the relevant university (ETH20-4734) to conduct secondary analyses on RTW with these variables.

### 2.7. Data Analysis

SPSS, version 26.0 was used to analyze the data. All data were checked for ambiguous coding prior to conducting the quantitative data analyses. Categorical variables were described as frequencies and percentages, and continuous variables as means and standard deviations (SD). The Chi-square or Fisher exact tests were used for the categorical variables, and Student *t*-tests were used for the continuous variables in the bivariate analyses to assess the relationships between the potential independent variables and RTW as the dependent variable. In the multivariate analyses, the associations between independent variables and RTW were assessed using binary logistic regression analysis, as RTW was considered a binary outcome. To ensure there was no strong correlation between the potential explanatory variables, a test for multicollinearity was applied using Kendall’s Tau. All results of the logistic analyses were presented as *p*-values and odds ratios (ORs) with 95% confidence intervals (CIs). Categories of the explanatory variables with the highest rates of RTW were used as the reference categories in the multivariate logistic regression analysis. Statistical significance was defined as *p* < 0.05 (two-tailed).

## 3. Results

### 3.1. Baseline Characteristics

Baseline characteristics were collected for the 136 participants within a few days after their admission to the hospitals with a diagnosis of MI. The mean age of the sample at baseline was 64.35 years (SD = 11.61), with a range of 21 to 86 years; the majority (*n* = 100, 73.5%) were male. Most were married (*n* = 119, 87.5%), and over half were educated to high school level or above (*n* = 79, 58.0%). Less than half the participants were employed at baseline (*n* = 56, 41.2%), and they mostly perceived their financial situation as “only fair” or “poor” (*n* = 118, 86.8%). More patients were diagnosed with non-ST elevation MI than ST elevation MI (*n* = 79, 58.1%), had experienced a previous MI (*n* = 106, 77.9%), received angioplasty (*n* = 112, 82.3%), and had a LVEF ≥ 40% (*n* = 119, 87.5%) at baseline. The demographic and clinical characteristics of the participants at baseline are presented in Table 1.

### 3.2. Demographic, Behavioral, Clinical, and Psychosocial Characteristics at Three Months Post-Discharge

At three months post-discharge, the number of participants who self-evaluated their financial status as “only fair” or “poor” had increased slightly by 2.1% (*n* = 121, 88.9%). More than half the participants were still cigarette smokers (*n* = 81, 59.6%). In addition, 13.2% (*n* = 18) had been readmitted to hospital, 12% (*n* = 16) had visited their doctor more than three times, and 14% (*n* = 19) experienced mild to severe chest pain most days. Regarding their psychological status, 23.5% (*n* = 32) of participants experienced mild to extremely severe depression, 14.7% (*n* = 20) experienced anxiety and 10.3% (*n* = 14) experienced stress over the follow-up period. The mean perceived social support scores based on the ESSI at three months after discharge was 27.97 (SD = 5.41), with the scores ranging between 11 and 34. The demographic, behavioral, clinical, and psychosocial characteristics displayed at a three-month follow-up are also shown in Table 1.

### 3.3. Return to Work at Three Months Follow-Up

Of the 136 participants, 56 (41.2%) were employed at the time of MI, while 80 (58.8%) did not have a job. At the three-month follow-up, 87.5% (*n* = 49) of the participants who were working at baseline had resumed work.

### 3.4. Predictors of Return to Work at Three Months Follow-Up

The characteristics of participants who did and did not RTW by three months post-discharge are compared in Table 2. Ten potential independent variables were selected on a theoretical basis using the findings of the systematic review conducted by Sun et al. [24]. The variables that were shown in previous studies to have statistically significant associations (*p* < 0.05) with an RTW post-MI were age, gender, educational level, smoking, LVEF, diabetes, hypertension, depression, anxiety, and social support [24]. Rates of an RTW were statistically significantly different in females vs. males. Patients who had not RTW at three months were more likely to be female, less educated, current cigarette and alcohol consumers, with diabetes and more comorbidities, and to have experienced readmission to a hospital after discharge (all *p* < 0.05). There were also statistically significant differences in RTW statuses based on the participants’ age, social support, anxiety, and depression scores. Older participants, those with less social support, and those with more severe symptoms of anxiety and depression were less likely to achieve work resumption by three months post-discharge (Table 2).

The results of the regression analysis showed that of the 10 variables entered into the regression model simultaneously, those including age, gender, smoking, anxiety, and depression were found to be independently significant predictors of an RTW in the study sample (Appendix A). That is, an older age, female gender, smoking, greater depression, and anxiety statistically significantly predicted a lower work resumption at three months follow-up post-discharge. The result of Kendall’s Tau rejected the existence of multicollinearity when checking the correlations between the independent variables.

The omnibus tests of the model coefficients were statistically significant (*p* < 0.001), meaning that the explained variance by the set of data was significantly greater than the unexplained variance. The Nagelkerke R Square was 0.558, which means that the independent variables included in the model explained 55% of the variance of an RTW at three months.

## 4. Discussion

In the current study, 87.5% of the participants who had a job at baseline had an RTW three months after discharge. Factors that were related to the patients’ work resumption in Korean patients with MI included age, gender, education, smoking, readmission after discharge, comorbidities (particularly diabetes), social support, anxiety, and depression. Males and those who were younger, did not smoke, and did not develop depression and anxiety, were more likely to RTW at three months post-discharge in the regression analysis. No previous study has reported RTW rates in patients with MI in South Korea, but the results differed from the available data of another study from Iran, which reported that 79.2% of participants who were employed prior to their MI had achieved work resumption at the three-month follow-up [35]. In another study from the Netherlands, only 46.2% of participants who had a paid job at baseline had resumed their previous work at the three-month follow-up [36].

The variation in rates of RTW may be, in part, attributable to the way an RTW was defined and reported. Methodological factors, such as varying RTW definitions, study samples, setting, and data collection methods, may affect the results. One of the reasons that the current study had the highest RTW rates among people with a job at baseline might be because the current study considered and counted self-employed people as holding a job and RTW, whilst studies conducted by Attarchi et al. [35] and de Jonge et al. [36] did not do so. Clinical factors may also contribute. A potential reason for the low RTW rate of 46.2% in the study by de Jonge et al. [36] might be that they used more rigorous criteria for diagnosing MI. These included meeting a minimum of two of the following criteria: 20 min of chest pain; elevated serum cardiac markers; and new pathological Q-waves on the electrocardiogram in at least two leads. As a result, they excluded cases of MI that other studies would have included. The current study did not have such specific inclusion criteria; provided patients had a diagnosis of MI by a cardiologist recorded in their medical record, they were considered eligible to participate. Consequently, the participants in this study may have demonstrated more minor symptoms of MI with a less acute condition during and after their hospital stay than in de Jonge’s et al. [36] study sample, eventually retaining a higher functionality to achieve work resumption. Furthermore, the data in the study by de Jonge et al. [36] were collected from 1997 to 2000, about 15 years prior to the current study data (collected from 2015 to 2016). It is likely that developments in MI treatment and care will have resulted in patients recovering with better functionality, thus increasing their capacity for resuming employment [9,13].

In the current study, patients who RTW at three months post-MI were more likely to be younger, and this finding is in line with previous research [12,36]. Older age has been frequently reported to predict poorer cardiovascular outcomes. Increased age was found to be associated with a higher risk of developing a major adverse cardiovascular event (MACE): recurrent MI, stroke, or cardiovascular death within one year post-MI [37,38]. This may suggest that MI at an older age is associated with more severe damage. Thus, older age may be considered a barrier for patients to RTW following MI, especially for those with a physically demanding job [36].

The findings from this study were consistent with most other studies in demonstrating that women were less likely than men to RTW after MI [8,9,11]. This finding may be explained by the fact that women with MI tend to be older at presentation and have poorer socio-economic status than males. For example, women are more likely to be single, work only part-time, or be unemployed, and where this is necessary to access healthcare, to lack health insurance [33,34,35,36]. They also tend to have poorer cardiovascular risk profiles, such as higher rates of hypertension, diabetes, heart failure, and chronic kidney disease, and to be less likely to receive optimum treatment for their MI, such as PCI revascularization procedures and referrals to CR [39,40,41,42]. Multiple studies have also shown that MI outcomes are poorer in women than men. For example, women had a higher likelihood of having MACE, depression, and anxiety, and worse general health and QOL after MI [39,41].

The gender difference in RTW post-MI may also be partially explained by the gender-based differences in the recovery goals set by women and men after MI. In the study by Grande and Romppel [43], female patients were more likely to maintain their mental and emotional health after MI and set goals around the ability to live independently and perform household tasks. For male patients, it was more important to achieve work resumption, decrease job strain, and increase physical endurance during their recovery from MI. This might reflect the traditional gender role expectations and affect the RTW decisions of both genders.

In the current study, smokers were less likely to RTW after MI than non-smokers or those who had quit smoking. The results of previous studies examining cardiovascular and other health outcomes among smokers and non-smokers after MI are mixed. In some studies, smoking was a protective factor for patients who experienced MI. For example, adverse left ventricular remodeling was less likely to occur in smokers [44], and they had a lower risk of experiencing MACE post-MI compared to non-smokers [45]. This has been referred to in the literature as the ‘smokers’ ‘paradox’ [44]. Rakowski et al. [46] argued that such findings were mainly due to the favorable baseline characteristics among smokers, such as being younger and with fewer comorbidities. In Rakowski’s et al. [46] study, although smokers had lower mortality rates than non-smokers post-MI in the univariable analyses, the difference between groups was not statistically significant after adjusting for age and gender.

Other studies observed that smokers had a worse prognosis than non-smokers post-MI. Compared to non-smokers and quitters, persistent smokers experienced poorer physical and mental health, QOL, higher cardiovascular and all-cause mortality, and had a greater risk of developing heart failure and MACE post-infarction [47,48,49].

In the current study, depression and anxiety were correlated with RTW outcomes in both bivariate and regression analyses. Patients who had depression and/or anxiety were less likely to have resumed work at the three-month follow-up. This finding is consistent with previous research [13,36,50]. Depression and anxiety are common comorbidities in patients with MI and correlate with a poor prognosis in this patient population. Patients with depression have higher risks of developing complications and MACE, and have a greater all-cause mortality post-MI [51,52]. Rates of recurrent MI were seen to be higher in those with anxiety [53]. While depression and anxiety impede an RTW post-MI, the symptoms of anxiety and depression improved in patients who did RTW [8]. However, these mental disorders often remain underdiagnosed and under-treated in this population; hospitalization periods are short, and some patients may develop depression and/or anxiety after discharge from the hospital [13,15].

In line with other studies, this study found that a higher educational level was positively associated with patients’ decisions to RTW after MI [10,12,13,36]. A higher educational level is usually an indicator of a higher-status occupation, greater income, and better socio-economic status. People with higher education are more likely to have white-collar jobs, facilitating an RTW post-MI, whereas people with less education often have physically demanding jobs with lower salaries, and as such, these jobs usually do not require higher qualifications [13,50]. Consequently, individuals may lack the physical capacity and/or motivation to return to labor-intensive roles after experiencing MI. Further, due to their education or training limitations, it may also be difficult for them to find an alternative, more sedentary position, thus leading to a worse socio-economic status [10,13].

Our findings share similar patterns to the results of other studies that examined the differences in RTW post-MI according to perceived social support, demonstrating higher RTW rates in patients with greater social support [10,54,55]. Social support is an important predictor of patient and disease outcomes after MI, and research has shown that patients with low social support have a higher risk of developing angina, have a worse QOL and physical and mental health status, and experience more depressive symptoms during the first year following MI [56,57]. Similarly, in a qualitative study, participants linked their ability to cope with their stressful situations, their adherence to their medication regimes, and an RTW to the social support that they received from family members, friends, and colleagues [54].

In the current study, patients with more comorbidities were less likely to resume work three months post-discharge, which is consistent with the findings of Dreyer et al. [10]. Diabetes was also found to be a statistically significant factor that related to a lower possibility of an RTW among these patients, a finding consistent with previous studies [9,10,12]. Additionally, patients with MI who were readmitted to a hospital after discharge were less likely to RTW within three months post-MI in the bivariate analysis, and this finding lends support to the study by Warraich et al. [9]. These factors may reflect patients’ severity of injury from their MI. Several studies reported comorbidities as associated with poorer clinical and recovery outcomes in patients with MI, such as a higher risk of in-hospital mortality, ICU admission, longer hospital stay, readmission within 30 days, and mortality at one-year post-MI [58,59,60]. Diabetes was associated with worse health status outcomes post-MI, such as readmission within 30 days, development of heart failure during hospitalization and in the year following MI, and higher all-cause mortality rates at five years post-MI [61,62]. A review reported that in patients with MI, angina, another episode of MI, and heart failure were the major reasons for their 30-day readmission, and common comorbid risk factors for 30-day readmission included renal dysfunction, heart failure, and diabetes [62]. This indicates that patients with comorbidities may have more adverse health statuses and require a more complex care plan to manage their conditions, including an RTW.

### 4.1. Implications

#### 4.1.1. Clinical Practice

The results of this study largely confirmed prior knowledge about an RTW after an MI, and emphasized the need for patient-centered approaches to decision-making for an RTW post-MI. Attention should be paid to the assessment of these patients’ mental health, and mental health care should be offered to people who require this, following MI. Any comorbidities experienced by patients with MI should not be neglected when caring for them. This information may help healthcare providers design and study different MI treatment and management practices to benefit patients with MI who have complex medical conditions. Study findings also emphasized the importance of identifying the reasons why patients with MI are readmitted to a hospital post-discharge. Solutions should be sought to reduce these patients’ risks of readmission post-discharge. The findings made clear the crucial clinical importance of encouraging smoking cessation for patients following MI. Family, friends, and social networks at the workplace, as social support, should be given suggestions by healthcare professionals on how to help patients return to normal (smoke-free) lives after an MI. The results of the study also underscored the need for healthcare providers to assess the risk factors after MI. This will help them identify patients with risk factors that deter work resumption so as to work with these patients, their family members, friends, and employers by targeting and addressing these modifiable factors to help them achieve this patient-centered outcome.

#### 4.1.2. Education for Healthcare Providers and Patients

Healthcare professionals require advanced education about health counseling services in order to prepare them to have the capacity to counsel and advise patients in relation to an RTW post-MI. Patients with MI, and their families, may wish to know when and to what extent they are physically safe and ready for an RTW post-MI. They are likely to require basic education regarding MI to increase their knowledge and understanding of this disease, and to better support informed and individualized decision-making on an RTW post-MI.

#### 4.1.3. Policy Making

Future health policies and guidelines should focus on addressing modifiable factors with strong relationships with an RTW post-MI, such as smoking, depression, anxiety, and social support. Likewise, as patients with multiple comorbidities may present differently and have a greater risk of adverse clinical outcomes than those with fewer comorbid conditions, clinical policies, and guidelines should differentiate between the less complex patients and those experiencing MI with multi-comorbidities. Clinical management policies or guidelines should be developed, specifically focusing on patients with MI and multiple cardiac or non-cardiac comorbidities to improve the in-hospital and long-term outcomes, including an RTW, of these high-risk patients. Additionally, considering that a significant proportion of the elderly Korean population remains in the labor market, policies regarding the creation of personalized work contracts should be introduced for these aging workers. Protected by such a policy, a flexible and selective work contract might be arranged between an employer and a Korean patient recovering from MI. Such a policy may allow Korean patients with MI to choose what time to start and finish their work during a workday, and their working hours might be reviewed and adjusted at intervals through negotiation between the employer and employee, as these patients are more vulnerable due to their age, illness, and reduced physical capacity.

#### 4.1.4. Future Research

Studies with a larger sample size and gender balance are required to examine the predictors of an RTW after MI. A comprehensive list of potential predictors should be included in further studies to improve the total variance explained. Researchers should consider using a consistent definition of RTW and objective measures of patients’ work statuses. Interventional studies should be considered to examine the causal relationships between these factors and patients’ RTW outcomes after MI in the future. Evidence is lacking to show if managing patients’ anxiety and depression after MI can indeed improve their RTW statuses. Further, there is a need to evaluate the effects of different treatment strategies for MI, including the provision of environments to support the RTW of patients post-MI.

### 4.2. Limitations

The results of this study should be interpreted in the context of several limitations. Using an existing dataset, some data related to the participants’ employment were missing, including their work type and salary, which have been shown to be significantly related to an RTW in patients with MI. Participants who did not work prior to their MI were included in the analyses, as we aimed to focus on their ability to work as a laborer rather than returning to their ‘previous’ jobs. Only paid work was considered as an RTW in the study, as including unpaid workers would confuse the results. Additionally, rather than categorizing employment as full- and part-time, data on RTW outcomes were collected as regular or casual work, with atypical terminologies as classifications of employment status. The gender distribution of the present cohort was not balanced, with women under-represented. The sample size (*n* = 136) was smaller than other studies that observed an RTW in people following MI, so the results should be interpreted with caution. The instruments used in this study relied on self-reporting, including the DASS-21, the ESSI and RTW status, and cigarette and alcohol consumption, which may have introduced bias to the study. In addition, according to the original study [22], patients with MI with deteriorating physical conditions were not willing to participate in the study, which could have introduced a selection bias. Given the observational nature of this study, inferences on causality cannot be made for these outcomes.

## 5. Conclusions

By collating evidence on the proportion of patients who RTW post-MI and the factors that predict work resumption, this study has generated new evidence on RTW statuses and predictive factors in patients with MI in South Korea. MI can be life-threatening and may require survivors to modify their long-term lifestyles. Empowering patients to take a central role in decision-making may optimize the treatment decisions made and patient recovery. An RTW post-MI is an important patient outcome and is beneficial for the health economy and individuals’ financial status, mental health, self-esteem, and ambitions.

The key findings from this study suggest that the majority of patients in work who experience MI have the capacity to achieve work resumption by three months post-discharge. When making new health policies or guidelines regarding CR for patients with MI, the timeframe of three months post-discharge can be considered a broadly appropriate time for patients with MI to RTW.

A younger age, male gender, not smoking, and not experiencing anxiety and depression, significantly predicted an RTW at three months post-discharge among Korean patients with MI. Study findings indicated factors for which interventions can be introduced to ameliorate the impacts of multiple modifiable factors in order to support an RTW post-MI. Such interventions should be incorporated within CR to become routine care for patients with MI during their recovery. The study findings indicated the importance of vocational evaluation and counseling services provided by healthcare professionals. Healthcare providers also need further education to support their delivery of new interventions to address factors that deter an RTW post-MI. Future research should include a health economic analysis to establish the return on investment and cost-effectiveness of specific interventions that target an RTW post-MI.

## Figures and Tables

**Figure 1 ijerph-19-08032-f001:**
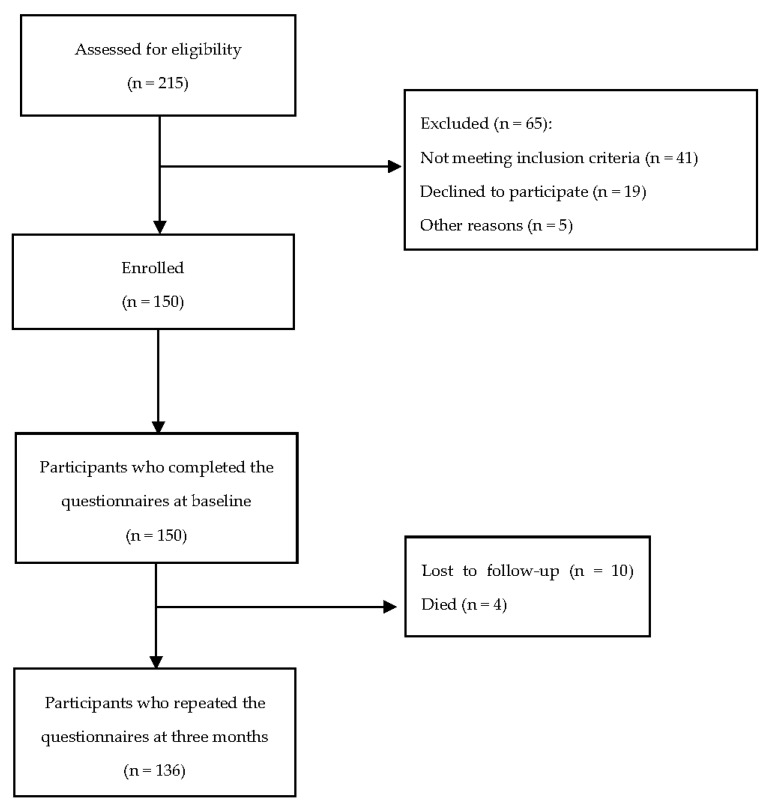
Flow diagram of participant recruitment.

**Table 1 ijerph-19-08032-t001:** Participants’ characteristics at baseline and three-month follow-up (*n* = 136).

BASELINE			FOLLOW-UP		
Variables	*n*	%	Variables	*n*	%
Demographic factors			Demographic factors		
** *Age* **			** *Self-rated financial status* **		
≤59	45	33.00	Excellent	2	1.50
60–69	41	30.10	Good	13	9.60
70–79	40	29.40	Only fair	92	67.60
80≤	10	7.50	Poor	29	21.30
** *Gender* **			**Behavioral factors**		
Male	100	73.50	** *Physical activity status* **		
Female	36	26.50	Moderate physical activity, at least 30 min most or all days of the week	49	36.00
** *Marriage status* **					
Married	119	87.50			
Never married/Separated/Divorced/Widowed	17	12.50	Moderate physical activity, less than 30 min less than 5 days per week	27	19.90
** *Educational level* **			Not physically active	60	44.10
Primary school	35	25.70	** *Smoking status* **		
Middle school	22	16.30	Smoker	81	59.60
High school	49	36.00	Non-smoker	26	19.10
Undergraduate study or more	30	22.00	Quit smoking	29	21.30
			** *Drinks alcohol* **		
** *Employment status* **			Yes	40	29.40
Regular employee	30	22.10	No	96	70.60
Casual employee	3	2.20	**Clinical factors**		
Self-employed	23	16.90	** *Readmission to hospital* **		
Unpaid family worker	24	17.60	Yes	18	13.20
Retired/unemployed	56	41.20	No	118	86.80
** *Self-rated financial status* **			** *Frequency of visiting a doctor within three months* **		
Excellent	3	2.20			
Good	15	11.00	Less or equal to 3 times	117	86.00
Only fair	82	60.30			
Poor	36	26.50	More than 3 times	16	12.00
**Clinical factors**			Unknown	3	2.00
** *Type of MI* **			** *Frequency of chest pain* **		
NSTEMI	79	58.10	No pain most days	117	86.00
STEMI	57	41.90	Mild pain most days	17	12.50
** *Time from MI to intervention* **			Moderate pain most days	2	1.50
Less than 2 h	50	36.80	**Psychosocial factors**		
More than 2 h	47	34.60	** *Depression scores* **		
Unknown	39	28.60	Not depressed	104	76.50
** *LVEF at admission* **			Mild/Moderate depression	26	19.10
≥55%	53	39.00	Severe/extremely severe depression	6	4.40
40–54%	66	48.50	** *Anxiety scores* **		
35–39%	8	5.90	No anxiety	116	85.30
<35%	9	6.60	Mild/moderate anxiety	15	11.00
** *Type of intervention* **			Severe/extremely severe anxiety	5	3.70
Medical antithrombotic therapy	20	14.70	** *Stress scores* **		
			Not stressed	122	89.70
Angioplasty	112	82.30	Mild/moderate stress	14	10.30
CABG	4	3.00	Severe/extremely severe stress	0	0
** *Number of comorbidities* **					
Zero	44	32.40			
One	47	34.60			
Two	41	30.10			
Three	4	2.90			
** *Previous MI* **					
Yes	106	77.90			
No	30	22.10			
** *Hypertension* **					
Yes	*66*	48.50			
No	*70*	51.50			
** *Diabetes* **					
Yes	41	30.10			
No	95	69.90			

**Abbreviations:****CABG:** coronary artery bypass graft; **LVEF:** left ventricular ejection fraction; **MI:** myocardial infarction; **NSTEMI:** non-ST elevation myocardial infarction; **STEMI:** ST elevation myocardial infarction.

**Table 2 ijerph-19-08032-t002:** Differences in characteristics of MI patients between those who were and were not working at 3 months (*n* = 136).

CATEGORICAL FACTORS					
Variables		3 Months Post Discharge (*n* = 136)			Chi-Square & *p*-Value
		RTW (*n* = 49)	NRTW (*n* = 87)		
**Demographic factors**					
** *Gender* **					Fisher’s exact *p* < 0.001
Female		3	33		
Male		46	54		
** *Marriage status* **					Fisher’s exact *p* = 0.110
Married		46	73		
Never married/Separated/Divorced/ Widowed		3	14		
** *Educational level* **					χ^2^ = 22.199 *p* < 0.001
Primary school		2	33		
Middle school		7	15		
High school		23	26		
Undergraduate study or more		17	13		
** *Self-rated financial status at 3 months* **					Fisher’s exact *p* = 0.001
Excellent/Good		13	5		
Only fair/Poor		36	82		
**Behavioral factors**					
** *Physical activity status at 3 months* **					χ^2^ = 5.668 *p* = 0.059
Moderate physical activity, at least 30 min		22	27		
most or all days of the week					
Moderate physical activity, less than 30 min		12	15		
less than 5 days per week					
Not physically active		15	45		
** *Smoking status at 3 months* **					χ^2^ = 13.494 *p* = 0.001
Never smoked		61	20		
Quit smoking		15	11		
Smoker		11	18		
** *Drinking alcohol at 3 months* **					χ^2^ = 14.127 *p* < 0.001
Yes		24	16		
No		25	71		
**Clinical factors**					
** *Diabetes* **					Fisher’s exact *p* = 0.011
Yes		41	54		
No		8	33		
** *Hypertension* **					Fisher’s exact *p* = 0.373
Yes		25	42		
No		21	45		
** *Number of comorbidities* **					χ^2^ = 7.543 *p* = 0.023
None		23	21		
One		14	33		
Two and more		12	33		
** *Previous MI* **					Fisher’s exact *p* = 0.052
Yes		43	63		
No		6	24		
** *Type of MI* **					Fisher’s exact *p* = 0.593
NSTEMI		30	49		
STEMI		19	38		
** *Time from MI to intervention* **					Fisher’s exact *p* = 0.437
Less than 2 h		24	26		
More than 2 h		20	27		
Unknown		5	34		
** *Type of intervention* **					χ^2^ = 0.885 *p* = 0.642
Medical antithrombotic therapy		41	70		
Angioplasty		2	2		
CABG		6	15		
** *Readmission to Hospital* **					Fisher’s exact *p* = 0.018
Yes		2	16		
No		47	71		
** *Frequencies of visiting a doctor within 3 months* **					Fisher’s exact *p* = 1.000
Less or equal to 3 times		43	74		
More than 3 times		6	10		
** *Frequency of chest pain within 3 months* **					Fisher’s exact *p* = 0.536
No/Mild pain most days		49	85		
Moderate/Severe pain most days		0	2		
**CONTINUOUS FACTORS**					
	**Have you returned to your normal work?**	**Mean**	**Standard deviation**	**t**	** *p* ** **value**
**Demographic factors**					
** *Age* **	No	68.33	11.452	5.985	*p* < 0.001
	Yes	57.27	8.015		
**Clinical factors**					
** *LVEF* **	No	49.8490	11.29764	−1.469	*p* = *0*.144
	Yes	52.6502	9.45999		
**Psychosocial factors**					
** *Social support at 3 months* **	No	27.11	5.850	−2.746	*p* = 0.007
	Yes	29.49	4.169		
** *Anxiety* **	No	2.10	2.215	2.042	*p* = 0.043
	Yes	1.31	2.133		
** *Stress* **	No	3.26	3.208	−0.331	*p* = 0.741
	Yes	3.45	2.973		
** *Depression* **	No	3.93	4.120	4.485	*p* < 0.001
	Yes	1.14	1.848		

**Abbreviations: CABG:** coronary artery bypass graft; **LVEF:** left ventricular ejection fraction; **MI:** myocardial infarction; **NRTW:** not return to work; **NSTEMI:** non-ST elevation myocardial infarction; **RTW:** return to work; **STEMI:** ST elevation myocardial infarction.

## Data Availability

The data presented in this study are available upon request from the corresponding author.

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
