# Peer review of "Predicting Return to Work Following Myocardial Infarction: A Prospective Longitudinal Cohort Study"

_ijerph, 2022, doi:10.3390/ijerph19138032_

Round 1
Reviewer 1 Report
Thank you for inviting me to review this interesting study. The report is well written and appears to be competently executed.
There were a couple of minor things which I thought could be clearer (e.g. the difference between excluded and not meeting the inclusion criteria)
Author Response
Thank you for your comment. We revised section 2.3 about participants for clarifying the difference between exclusion and not meeting the inclusion criteria. Further, we examined the whole manuscript to provide more clarity.

Reviewer 2 Report
This study has elucidated new evidence on RTW status and predictive factors in patients with MI in South Korea. The objectives and study design are definite in line with the STROBE, and the results are carefully discussed. Although the sample size is limited, the study is highly valid.
Comments
p.3 Line 98-106
Regarding the novelty that previous literature from western countries has shown various findings on patients’ RTW following MI, but no study has focused on Korean population,
- what is the significance of targeting Korean population, i.e., did you anticipate results that differed from preceding research findings?
- what hypotheses, if any, would you have made?
If these are mentioned, the position of this study concerning previous studies will become more apparent. Also, if the results indicate implementing interventions unique to Korean situations, please highlight them.
Author Response
Thank you for your constructive comments. The manuscript has been enhanced accordingly. Please kindly find the attachment.

Reviewer 3 Report
The study is of interest in ​​occupational health and underlying factors that could be of interest to improve the quality of life of patients and to strengthen prevention and health promotion programs.
Regarding the design, it is not clear if it is a cross-sectional study and not necessarily a longitudinal study for the following reason
The researchers indicate that they will work with secondary data that evaluated the quality of life of patients after an MI, including clinical, behavioral, and psychosocial factors. While the current study focuses on the Return to Work after an MI. In this regard, for a design to be longitudinal, it must have at least two or more measurements of the variables that are of interest to measure. It is unknown if the researchers have three measurements, because otherwise the study would be cross-sectional.
On the other hand, it is stated that the sample obtained is taken from a study by Mayor Tamayo. In addition, inclusion criteria are applied that reduce the sample further, 'therefore, the study could have low statistical power. Even in the limitations it is commented that the sample was small, and this could affect the results.
It is important for researchers to clarify the issue of the design and the statistical power of the sample.
Author Response

(The authors gave the same response as above.)

Reviewer 4 Report
Recommendations for the authors of the article:
1. In the article, add and extract the section “Review of Literature. ” Particular emphasis should be placed on the importance of quality of life concepts.
2. It is appropriate to clarify the basis for the choice of variables for the second objective of the Article.
3. The article does not explain the importance of behavioural factors as research variables. Especially since the authors highlight them for the purpose of the article.
4. The section “Conclusions” should definitely be expanded. The conclusions of the studies shall be presented in sections.
Author Response

(The authors gave the same response as above.)

Reviewer 5 Report
The presented manuscript provides a post-analysis of a cohort study. in this paper, the effect of various factors on "return to work" after severe heart condition and hospitalization has been investigated.
There are comments which can be considered:
1- In some parts of the manuscript, the extended form of abbreviations for the first time has been missed (lines 50, 57,124,185,203, 243).
2- In subtitle 2.3, at the end of this paragraph the criteria for employed and unemployed participants need to be clarified, which is mentioned in subtitle 2.5.1.
3- In line 143, what the authors mean by "baseline data" is better to be clarified here for the first time in the context.
4- In title 2.6 the ethical code is better to be mentioned exactly.
5- Title 2.7 the statistical analysis should be summarized and only focus on statistical tests and other sentences would be better moved to the result section.
6- Table 3 can be moved to supplementary files.
7- The range of age in this study was up to 81. Do elders of this age work? Would not be better to assess people in the routine working-age such as 18 to 60? It seems the condition for employment in this study is wide but “return to work” is a phrase that reminds come back to workshops or offices. Please explain.
Author Response

(The authors gave the same response as above.)
